# VEGFR3 modulates brain microvessel branching in a mouse model of 22q11.2 deletion syndrome

Sara Cioffi[1], Gemma Flore[1], Stefania Martucciello[2], Marchesa Bilio[1], Maria Giuseppina Turturo[1], Elizabeth Illingworth[2]

The loss of a single copy of *TBX1* accounts for most of the clinical signs and symptoms of 22q11.2 deletion syndrome, a common genetic disorder that is characterized by multiple congenital anomalies and brain-related clinical problems, some of which likely have vascular origins. *Tbx1* mutant mice have brain vascular anomalies, thus making them a useful model to gain insights into the human disease. Here, we found that the main morphogenetic function of TBX1 in the mouse brain is to suppress vessel branching morphogenesis through regulation of *Vegfr3*. We demonstrate that inactivating *Vegfr3* in the *Tbx1* expression domain on a *Tbx1* mutant background enhances brain vessel branching and filopodia formation, whereas increasing *Vegfr3* expression in this domain fully rescued these phenotypes. Similar results were obtained using an in vitro model of endothelial tubulogenesis. Overall, the results of this study provide genetic evidence that *VEGFR3* is a regulator of early vessel branching and filopodia formation in the mouse brain and is a likely mediator of the brain vascular phenotype caused by *Tbx1* loss of function.

## Introduction

Seminal studies in rodents have described the process by which the brain is vascularized through angiogenic sprouting from the perineural vascular plexus (PNVP). In the embryonic mouse brain, vascularization initiates in the hindbrain around E10 and procedes in a caudal-rostral direction. Within the brain parenchyma, the number of blood vessels increases rapidly to form a dense vascular network that is able to support neurogenesis, which in the mouse initiates around E11.5.

*Tbx1* mutant mice are a model of 22q11.2 deletion syndrome (22q11.2DS), a relatively common chromosome microdeletion disorder, for which most of the clinical problems result from *TBX1* haploinsufficiency (Yagi et al, 2003; Paylor et al, 2006; Torres-Juan et al, 2007; Ogata et al, 2014). In practical terms, the most vexing aspect of the clinical phenotype is the brain-related clinical problems, some of which might have vascular origins. TBX1 is a member of the family of T-box transcription factors that is widely expressed in the

embryonic pharynx, and it is also expressed in the endothelial lining of a subset of brain vessels in pre-term mouse embryos (Paylor et al, 2006). We have previously shown that TBX1 plays a critical cell-autonomous role in brain endothelial cells (ECs) that determines the correct density and functionality of brain vessels (Cioffi et al, 2014). We have proposed that it exerts these effects through interactions with *Vegfr3* and *Dll4* in ECs, both of which have anti-angiogenic functions in the mouse brain (Suchting et al, 2007; Tammela et al, 2011) and are regulated by TBX1 in ECs, including brain ECs, in vivo and in vitro (Chen et al, 2010; Cioffi et al, 2014). Endothelial-specific inactivation of each of these genes causes brain vessel hyperbranching (Suchting et al, 2007; Tammela et al, 2011; Cioffi et al, 2014). We have previously shown that the Notch agonist JAG1 is not sufficient to rescue fully endothelial microtubule hyperbranching in cultured ECs silenced for *TBX1* (Cioffi et al, 2014), suggesting that *Vegfr3* and/or Notch-independent pathways are involved.

In this study, we demonstrate that *Tbx1* is activated in mature ECs, rather than in their mesodermal precursors, during early brain vascularization. Activation of expression initiates in blood vessels of the hindbrain neuroepithelium at E10.5 and then spreads to most of the brain by E15.5. We found a broad overlap between TBX1 and VEGFR3 in brain, vessels and importantly, genetic experiments determined that *Tbx1*-driven enhanced expression of *Vegfr3* is sufficient to rescue the brain vasculature phenotype of *Tbx1* mutants. Overall, our results indicate that VEGFR3 is a key mediator of the modulatory function of TBX1 in brain vessel branching.

## Results

### *Tbx1* expression is activated in ECs rather than in EC progenitors

Many of the TBX1-dependent functions in the cardiovascular system are linked to progenitors rather than to differentiated cells. As *Tbx1* is widely expressed in the head mesenchyme, we asked whether it is activated in EC progenitors in this tissue before they populate the brain. To this end, we crossed *Tbx1^{Cre/+}* mice with *Rosa^{mTmG}* reporter mice (Muzumdar et al, 2007) and evaluated the distribution of *Tbx1*-expressing cells and their descendants (lineage tracing) in serial coronal brain sections of *Tbx1^{Cre/+}; Rosa^{mTmG}*

---

[1]Institute of Genetics and Biophysics "ABT," CNR, Naples, Italy   [2]Department of Chemistry and Biology, University of Salerno, Fisciano, Italy

Correspondence: eillingworth@unisa.it

embryos between stages E9.5 and E18.5, where these cells are marked by *Tbx1^Cre*-activated expression of GFP. Sections were co-immunostained with anti-KDR, also known as VEGFR2 (E9.5–E11.5) or anti-GLUT1 (E13.5–E18.5) to identify ECs. Overall, results showed that *Tbx1*-expressing cells and their progeny (GFP+) populate the developing brain following caudal to rostral and ventral to dorsal trajectories, as shown in the cartoon in Fig 2M. Specifically, at E9.5 and E10.5, only a few ECs in the perineural vascular plexus (PNVP) were GFP+;KDR+, whereas the surrounding head mesenchyme was heavily populated with GFP+; KDR– cells (Fig 1A). The first GFP+ cells localized to blood vessels within the hindbrain neuroepithelium at E10–E10.5 (Fig 1B) in the form of isolated KDR+; GFP+ cells (arrows in

Fig 1B). Between E10.5 and E11.5, the number of GFP+ cells increased dramatically in the hindbrain neuroepithelium, such that by E11.5, co-expression of GFP and KDR was almost 100% (Fig 1C). However, at this stage, the midbrain (Fig 1D) and forebrain (Fig 1E) contained only a few isolated GFP+;KDR+ cells. From E13.5 onward, we used the anti-GLUT1 antibody to label blood vessels. At E13.5 (Fig 2A–F), the presence GFP+;GLUT1+ cells increased in the ventral forebrain (Fig 2ab' and cd') and ventral midbrain (Fig 2ef'), but the dorsal forebrain (Fig 2ab and cd) and midbrain (Fig 2ef) remained largely free of GFP+ cells. The contribution to and distribution of GFP+ cells in the aforementioned brain structures were maintained at E15.5 (Fig 2G–L) in the forebrain (boxed areas in Fig 2G and I) and in the

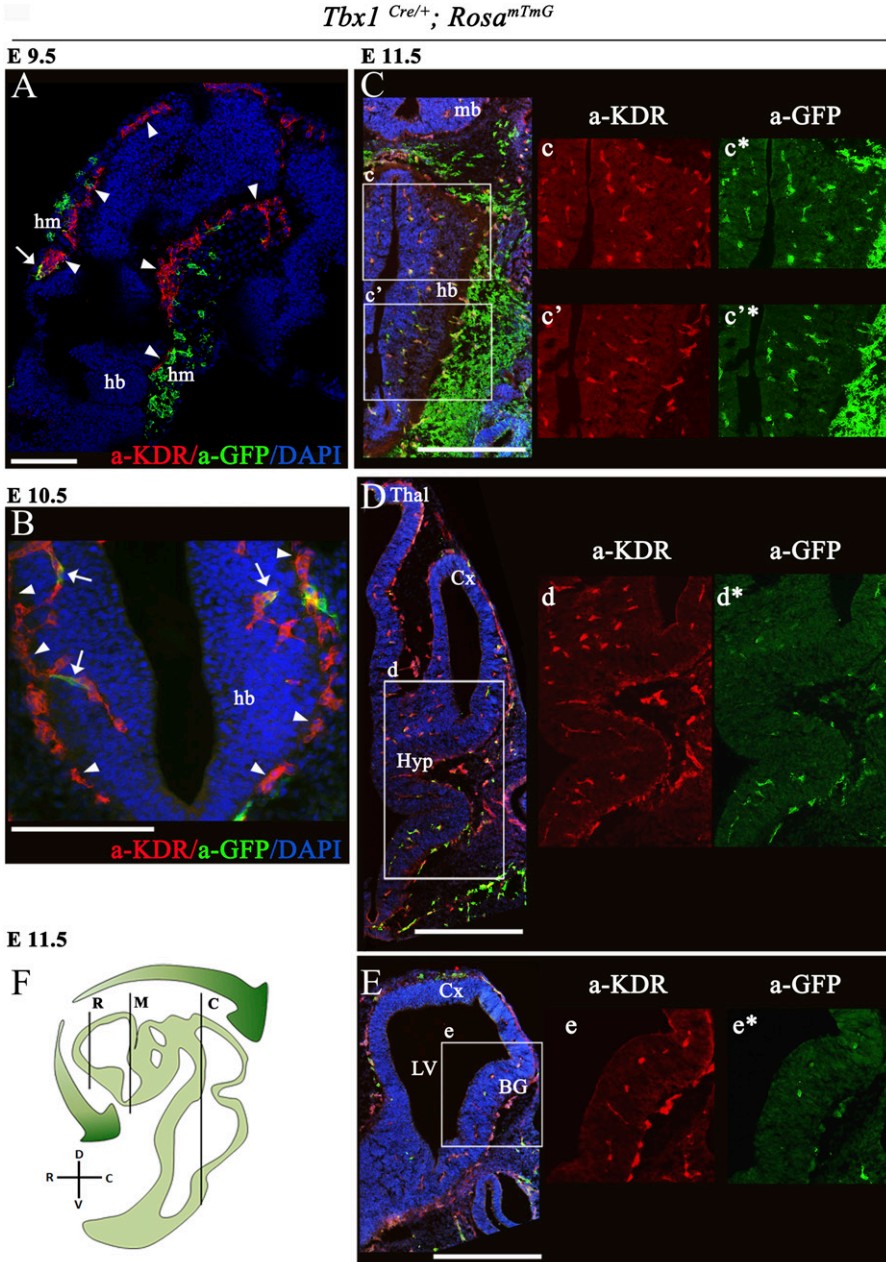

**Figure 1.  Distribution of *Tbx1*-fated cells in the brain of *Tbx1^Cre/+; Rosa^mTmG* embryos.**

**(A, B, C, D, E)** Representative transverse brain sections of embryos at E9.5 (A) and coronal sections at E10.5 (B) and E11.5 (C, caudal section), (D, medial section), and (E, rostral section) immunostained for GFP (green), which labels *Tbx1*-fated cells and for endothelial KDR (red). Nuclei (blue) were counterstained with DAPI. White arrowheads in (A) and (B) indicate the perineural vascular plexus and white arrows cells co-expressing GFP and KDR. White boxes in (C, D, E) (merge) are shown as enlarged, single-color channels to the right. **(C, D, E, F)** Cartoon shows the position of the coronal brain sections shown in panels (C) (caudal, C), (D) (medial, M), and (E) (rostral, R). Green-shaded arrows indicate the relative density of GFP+ cells (*Tbx1*-fated) along the rostral–> caudal brain axis at E11.5. Scale bars, (A) 250 μm, (B, C, D, E) 500 μm. Abbreviations: hm, head mesenchyme; hb, hindbrain; Cx, cortex; LV, lateral ventricle; BG, basal ganglia; mb, midbrain; Thal, thalamus; Hyp, hypothalamus.

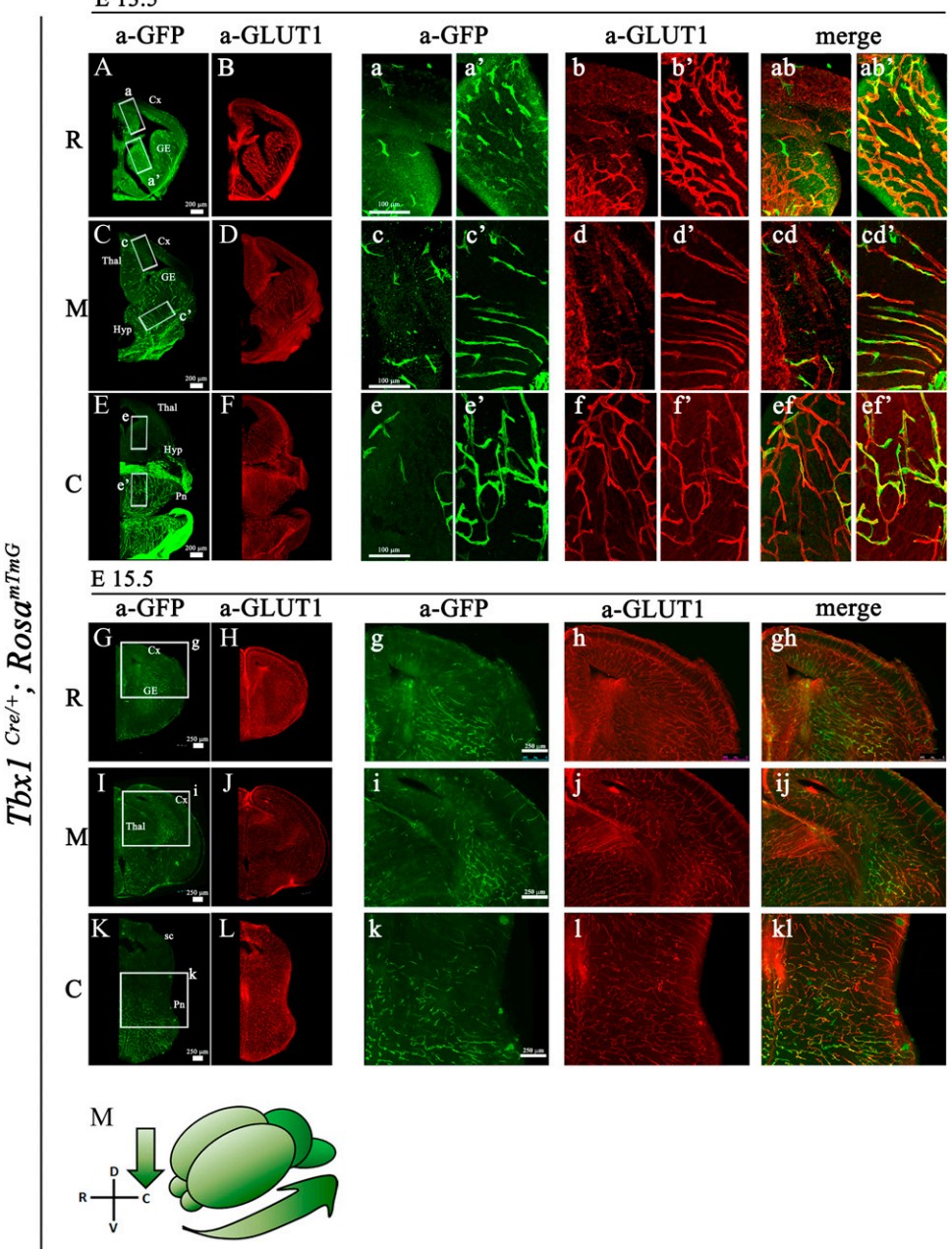

**Figure 2. Distribution of *Tbx1*-fated cells in the brain of *Tbx1^Cre/+^; Rosa^mTmG^ embryos. (A, B, C, D, E, F, G, H, I, J, K, L)** Representative coronal brain sections of embryos at E13.5 and (G, H, I, J, K, L) at E15.5 stained for GFP (green) and endothelial GLUT1 (red). Boxed areas in (A, C, E, G, I, K) are enlarged in the adjacent panels. **(M)** Cartoon showing the relative density of GFP+ cells (*Tbx1*-fated) along the rostral–> caudal and dorsal–> ventral axes. Abbreviations: R, rostral; M, medial; C, caudal; Cx, cortex; GE, ganglionic eminences; Thal, thalamus; Hyp, hypothalamus; Pn, pons; sc, superior colliculus.

midbrain (boxed area Fig 2K) and at E18.5 (Fig S1A–F), suggesting that *Tbx1* is not activated significantly in dorsal brain regions during embryonic development (Fig S1A, C, and E). At no developmental stage were GFP+/KDR-negative or GFP+/GLUT1-negative cells seen within the brain parenchyma, indicating that *Tbx1*-fated cells are ECs.

As brain vascularization progresses, newly formed vessels recruit extracellular matrix, mural cells, and pericytes that help to stabilize them. Pericytes and ECs have diverse origins that include progenitors in the paraxial mesoderm. We asked whether *Tbx1* is activated in mature pericytes or in their precursors. We tested this by co-immunostaining brain sections of *Tbx1^Cre/+^; Rosa^mTmG^* embryos with antibodies to PDGFRβ (labels pericytes), GLUT1, and GFP. However, high-magnification microscopy failed to distinguish EC-derived and pericyte-derived signals (Fig S2). Thus, we are unable to determine whether the *Tbx1* lineage includes pericytes.

### Endothelial tip cells are potential sites of TBX1 and *Vegfr3* interaction

To identify sites of TBX1-*Vegfr3* interaction, we first determined whether the anti-VEGFR3 antibody used in this study was pan-endothelial in the mouse brain as previously reported (Watanabe

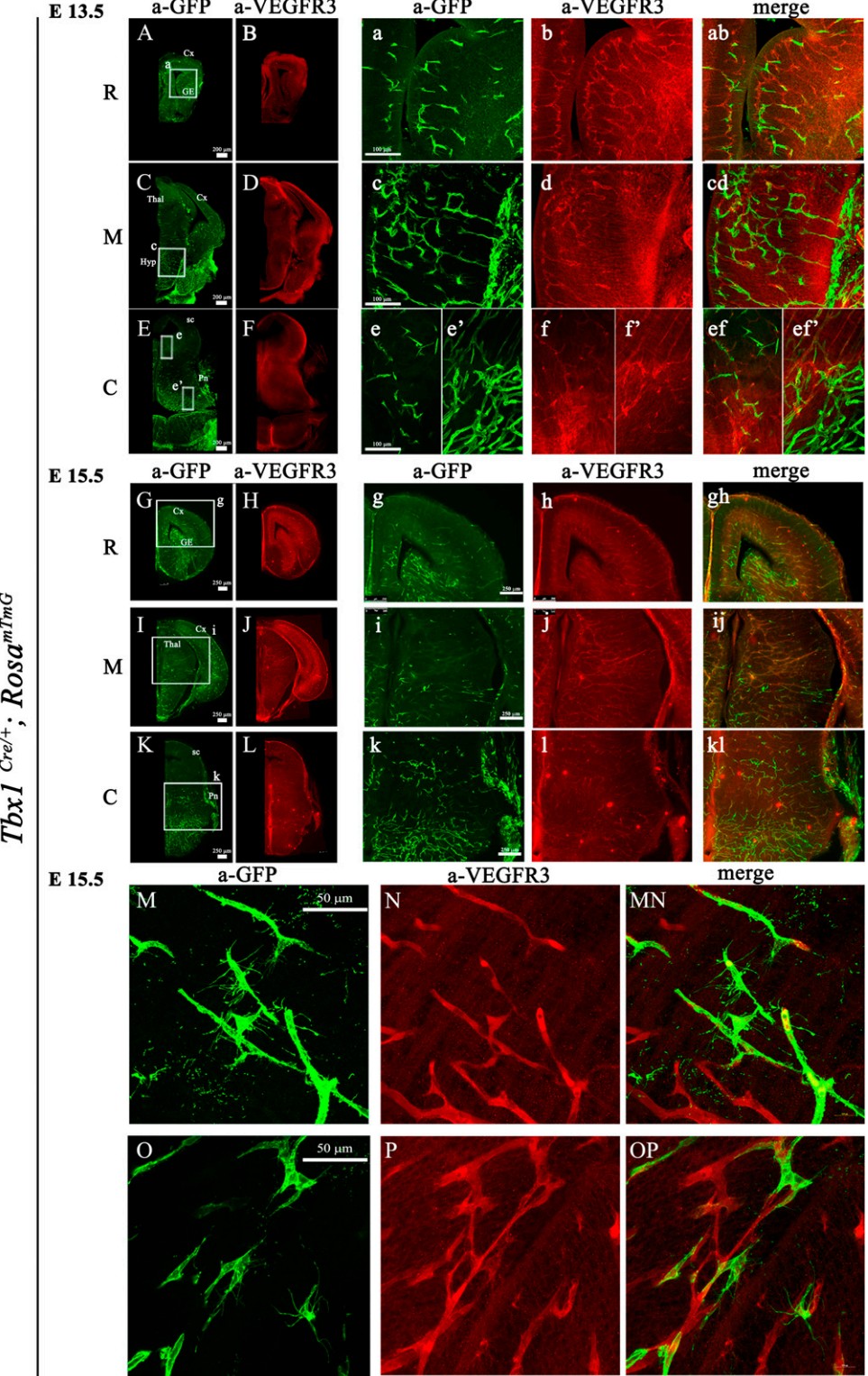

**Figure 3. TBX1-GFP and VEGFR3 co-localize in brain endothelial cells.**
**(A, B, C, D, E, F, G, H, I, J, K, L, M, N, O, P)** Representative coronal brain sections of *Tbx1$^{Cre/+}$; Rosa$^{mTmG}$* embryos at E13.5 and (G, H, I, J, K, L, M, N, O, P) at E15.5 immunostained for GFP (green) and endothelial VEGFR3 (red). Boxed areas in (A, C, E, G, I, K) are enlarged in the adjacent panels. **(M, N, O, P)** High-magnification images show that endothelial filopodia are labeled by anti-GFP (M, O) but not anti-VEGFR3 (N, P, MN, OP) antibodies. Abbreviations: R, rostral; M, medial; C, caudal; Cx, cortex; GE, ganglionic eminences; Thal, thalamus; Hyp, hypothalamus; Pn, pons; sc, superior colliculus.

et al, 2019). For this, and for the subsequent analysis of *Tbx1$^{Cre}$*-activated GFP and VEGFR3 localization, we used the same sectional series of *Tbx1$^{Cre/+}$; Rosa$^{mTmG}$* embryos. The GFP expressed by this reporter is targeted to the cytoplasmic membrane (Muzumdar et al, 2007). Co-immunostaining with anti-GLUT1 and anti-VEGFR3 antibodies revealed expression of the two proteins in virtually all brain

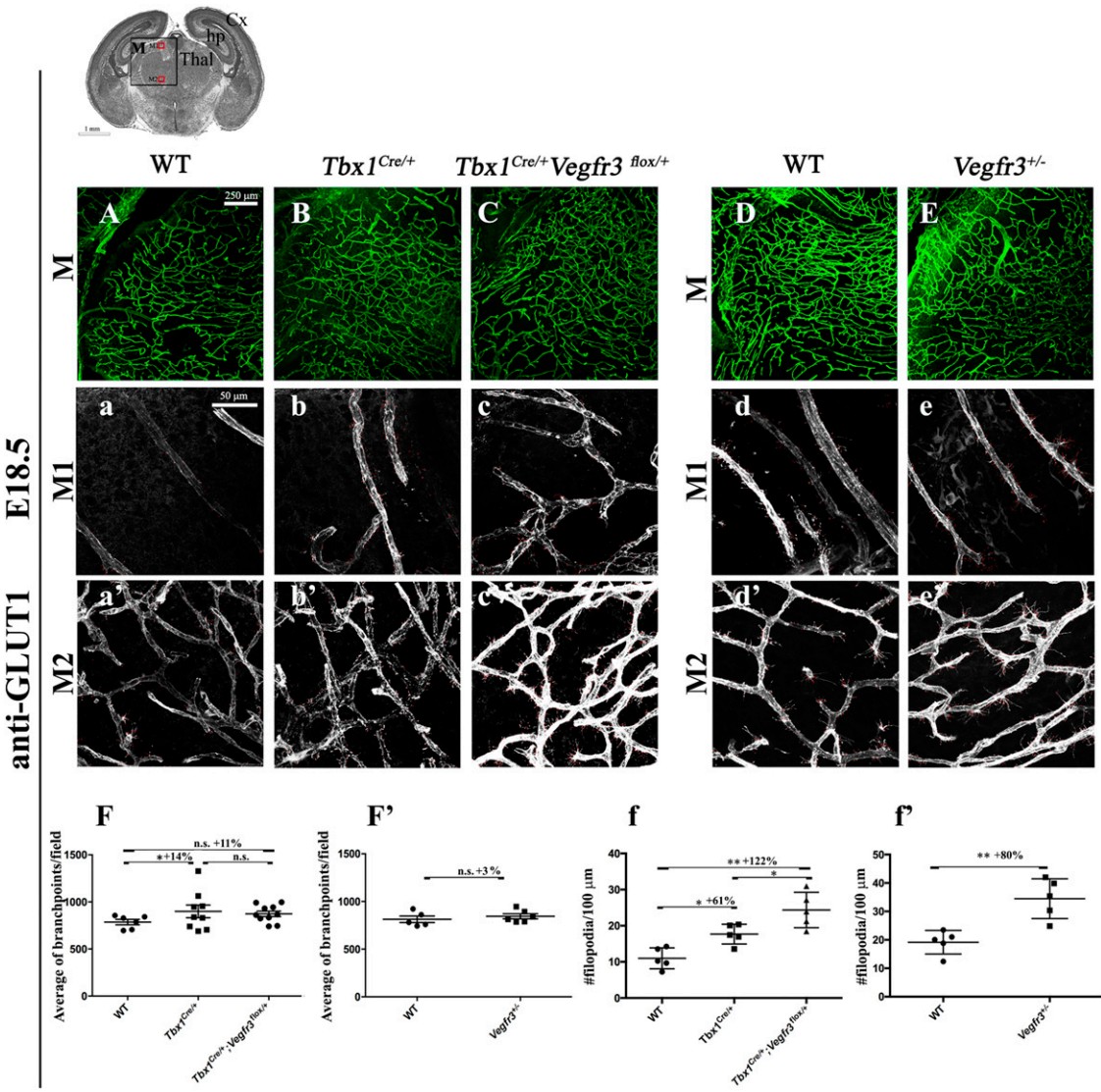

**Figure 4. *Tbx1* and *Vegfr3* interact genetically to regulate brain vessel and filopodial density.**
**(A, B, C, D, E)** Representative coronal brain sections (medial) of E18.5 embryos immunostained for GLUT1 (green). **(F, F')** Cartoon indicates the position of the counting boxes (M1, M2), shown at high magnification in panels M1 (a–e) and M2 (a'–e'), used to quantify vessel branch point (F, F') and filopodial (f, f') density in embryos with the indicated genotypes. ***P*-value < 0.001, ***P*-value < 0.01, **P*-value < 0.05. Error bars ± SD. Abbreviations: Cx, cortex; hp, hippocampus; Thal, thalamus.

vessels along the rostral-caudal brain axis at E13.5 (Fig S1G–L). We next co-immunostained an adjacent series of sections with anti-GFP and anti-VEGFR3 antibodies. This revealed extensive co-localization of the GFP and VEGFR3 proteins in brain vessels along the rostral-caudal brain axis at the stages analyzed, namely, E13.5 (Fig 3A–F, ab, cd, ef, and ef'), E15.5 (Fig 3G–L, gh, ij, and kl), and E18.5 (Fig S1M–R) including endothelial tip cells, which are distinguished from stalk cells by the presence of filopodia (Fig 3M–P). Many but not all GFP+ tip cells expressed VEGFR3, but two different anti-VEGFR3 antibodies failed to label the filopodia (Fig 3N and P). This may be because of an inability of the antibodies to bind to the receptor in this territory or to a lack of the receptor at this location in these mutants. Thus, GFP+; VEGFR3+ ECs, including angiogenic tip cells, are potential sites of TBX1-*Vegfr3* interaction.

## *Vegfr3* haploinsufficiency enhances the angiogenic sprouting phenotype of *Tbx1* mutants

We next tested whether TBX1 and *Vegfr3* interact genetically. For this, we intercrossed *Tbx1*[Cre/+] (*Tbx1*[Cre] is a null allele) and *Vegfr3*[flox/+] mice and quantified brain vessel branch point density and angiogenic sprouting (filopodial density) in thick (100 µm) brain cryosections of conditional compound heterozygous embryos at E18.5 immunostained with anti-GLUT1 (Figs 4A–C and S3). Results revealed an increased density of endothelial filopodia in *Tbx1*[Cre/+];*Vegfr3*[flox/+] embryos compared with *Tbx1*[Cre/+] embryos (Fig 4B, C, and f, *P* = 0.0317 or WT controls, *P* = 0.0079) but no differences in branch point density (Fig 4F). Thus, *Vegfr3* heterozygosity enhances the brain vasculature phenotype in *Tbx1* heterozygous (*Tbx1*[Cre/+])

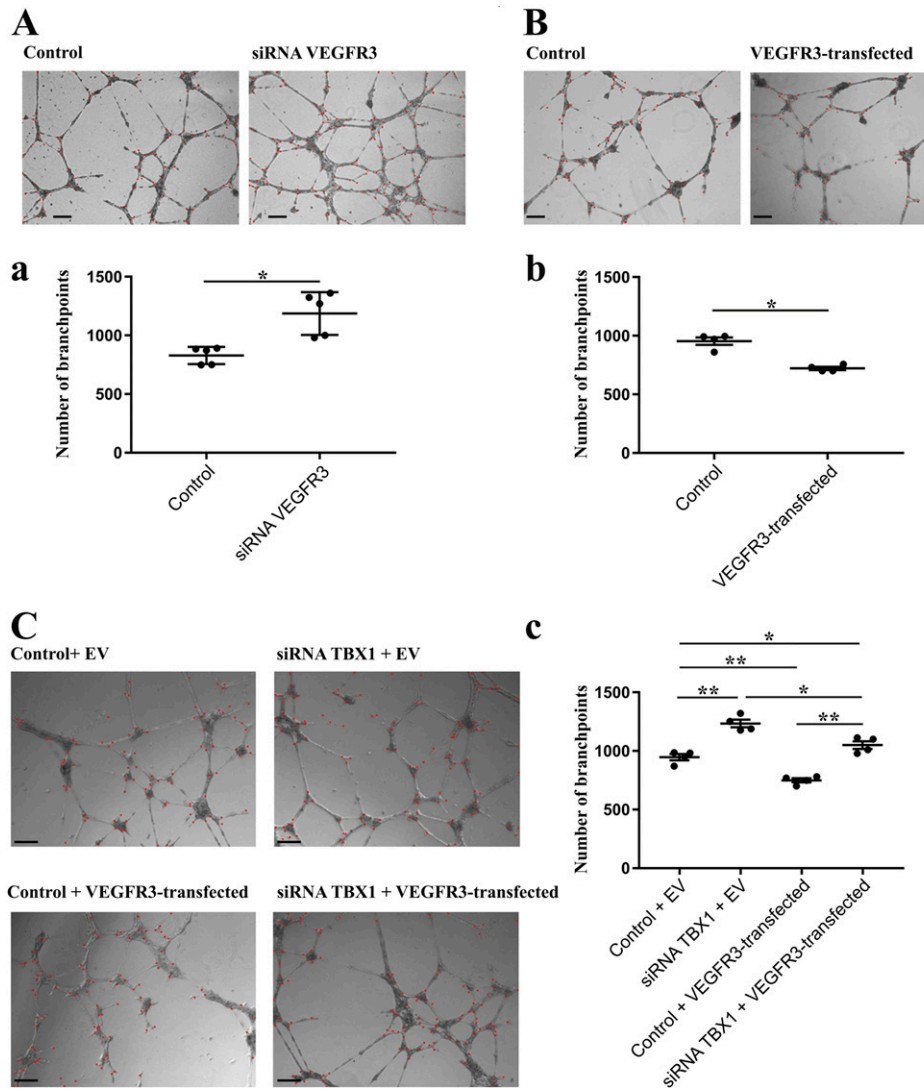

**Figure 5.  *Vegfr3* overexpression in HUVECs represses formation of endothelial microtubules.**
**(A)** Microtubule hypobranching after transfection with pcDNA3-*Vegfr3*.
**(B)** Hyperbranching after RNA interference, siRNA VEGFR3. **(C)** pcDNA3-*Vegfr3* rescued hyperbranching caused by *TBX1* knockdown. ***P*-value < 0.001, ***P*-value < 0.01, **P*-value < 0.05. Error bars ± SD. Scale bar, 200 µm. Abbreviations: EV, empty vector.

mutants, albeit mildly. Interestingly, a similar result was obtained in *Vegfr3*$^{+/-}$ (Fig 4E) embryos (germline heterozygous) versus WT embryos (Fig 4F' and f'), that is, increased filopodial density (*P* = 0.0079) but unaltered branch point density (*P* = 0.117), suggesting that there might be nonproductive angiogenic sprouting in *Vegfr3*$^{+/-}$ embryos.

The extensive co-localization of Tbx1-GFP and VEGFR3 in brain vessels, together with the predominantly ventral distribution of *Tbx1*-fated cells (GFP+), suggests that ventral rather than dorsal brain structures may be the preferred sites of TBX1-*Vegfr3* interaction and in particular, the ventral hindbrain, where *Tbx1* is activated in 100% of brain vessels by E11.5. VEGFR3 was expressed in most brain vessels already at E13.5 (this study and Watanabe et al [2019]), but brain vessel density was normal in *Tbx1*$^{-/-}$ embryos at this stage (*P* = 0.07, not shown), suggesting that the critical interaction between the two genes likely occurs between E13.5 and E15.5.

## Enhanced expression of *Vegfr3* in *Tbx1*-depleted ECs suppresses microtubule hyperbranching

Because reduced dosage of *Vegfr3* enhances the *Tbx1* mutant phenotype, we reasoned that VEGFR3 may be a mediator of TBX1 in regulating brain vessel density. If this were the case, supplemental *Vegfr3* expression should rescue the phenotypic consequences of *Tbx1* mutation. The likelihood of the success of this approach depends upon the effects of altered *Vegfr3* dosage on EC proliferation and vessel growth, but in vivo studies indicate that VEGFR3 has both pro- and antiangiogenic potential (Karkkainen et al, 2000, 2001; Mäkinen et al, 2001). Therefore, before initiating in vivo rescue experiments, we first tested in vitro whether ECs respond to *Vegfr3* overexpression in a way that would predict in vivo rescue. We tested this in a functional assay that contains only ECs (HUVECs) cultured in Matrigel (Fig 5). We have shown previously that *TBX1* knockdown in HUVECs causes microtubule hyperbranching in Matrigel

endothelial cultures (Cioffi et al, 2014). Thus, results obtained in the cell-based assay were consistent with in vivo data obtained in the same study in *Tbx1* mutant mice, which showed brain vessel hyperplasia (Cioffi et al, 2014). In Matrigel assays, we found that knockdown of *VEGFR3* by transfection with a *VEGFR3* targeting siRNA led to an ~60% increase in microtubule branching compared with HUVECs transfected with a control siRNA (Fig 5A, *P* = 0.0138), whereas overexpression of *Vegfr3* in HUVECs, achieved by transfection with a plasmid containing a full-length murine *Vegfr3* cDNA (pcDNA3-Vegfr3), led to ~24% reduced microtubule branching compared with HUVECs transfected with the empty plasmid vector (Fig 5B, *P* = 0.0027). We then tested whether overexpression of *Vegfr3* would reduce microtubule hyperbranching after *TBX1* knockdown. For this, we co-transfected HUVECs with pcDNA3-*Vegfr3* and siRNA-*TBX1* or siRNA-CTR (control) and measured microtubule branching 12 h after plating in Matrigel and 24 h after the second transfection. Results showed that overexpression of *Vegfr3* suppressed microtubule hyperbranching in HUVECs after *TBX1* knockdown, although branching did not return to wild-type levels (Fig 5C, *P* = 0.0148).

We asked whether TBX1 and VEGFR3 negatively regulate a common downstream target that might explain their apparently similar loss of function phenotypes. *Vegfr2* (*Kdr*) encodes a potent pro-angiogenic factor that is negatively regulated by TBX1 in various contexts (Cioffi et al, 2014; Lania et al, 2015; Martucciello et al, 2020) and is up-regulated after EC-conditional deletion of *Vegfr3* in neonatal mice (Zarkada et al, 2015). We tested whether *Vegfr2* dysregulation occurs in mouse brain ECs (bEND5; Sigma-Aldrich) in response to changes in *Tbx1* and *Vegfr3* gene dosage. We found that overexpression of *Tbx1* but not *Vegfr3* in these cells was associated with a modest but significant down-regulation of *Vegfr2* expression (Fig S4A and B). Analogously, in HUVECs, *VEGFR2* expression increased after *TBX1* knockdown and returned to control levels of expression in cells co-transfected with *TBX1* siRNA and pcDNA3-*Vegfr3* (Fig S4C). Thus, *VEGFR2* dysregulation might contribute to microtubule hyperbranching in HUVECs after *TBX1* knockdown.

### Transgenic expression of *Vegfr3* rescues brain vessel hyperplasia caused by loss of *Tbx1* function

We next asked whether overexpression of *Vegfr3* in the *Tbx1* expression domain would rescue the brain vessel abnormalities observed in *Tbx1* mutant embryos. To this end, we crossed *Tbx1*$^{Cre/+}$ mice with mice carrying a single copy of Cre-activatable murine *Vegfr3* transgene, TgVegfr3 (Martucciello et al, 2020). This transgenic line was selected to minimize the risk of producing high levels of VEGFR3 expression that might cause embryonic lethality. Immunostaining with anti-VEGFR3 and anti-GFP antibodies on brain sections of *Tbx1*$^{Cre/+}$;*Rosa*$^{mTmG}$ versus *Tbx1*$^{Cre/+}$;*Rosa*$^{mTmG}$;TgVegfr3 embryos at E18.5 (n = 3 per genotype) was not sufficient to reveal differences in VEGFR3 signal intensity between the two genotypes (Fig S5A, D, G, and J), but increased expression of *Vegfr3* transcripts was evident by qRT–PCR normalized for PECAM1, *P* = 0.009, and GAPDH, *P* = 0.01 (Fig S5M). We then intercrossed *Tbx1*$^{Cre/+}$;TgVegfr3 and *Tbx1*$^{lacZ/+}$ mice and measured brain vessel density and filopodial density in E18.5 embryos with the following genotypes: *Tbx1*$^{+/+}$, *Tbx1*$^{Cre/+}$, *Tbx1*$^{Cre/lacZ}$, TgVegfr3;*Tbx1*$^{Cre/+}$, and TgVegfr3;*Tbx1*$^{Cre/lacZ}$ that were immunostained with anti-GLUT1 (Figs 6 and S6). Results

revealed the presence of brain vessel hyperplasia (+17%, *P* = 0.0365) and increased filopodial density (+85%, *P* = 0.0079) in *Tbx1* heterozygous (*Tbx1*$^{Cre/+}$) embryos (Fig 6B–b', F, and f) and in *Tbx1* homozygous embryos (+35.5% branch points, *P* = 2.126 × 10$^{-9}$ and +58% filopodia, *P* = 0.0079, respectively) (Fig 6D–d', F, and f). More importantly, *Tbx1*$^{Cre/+}$-induced activation of the *Vegfr3* transgene fully rescued both brain vessel phenotypes in *Tbx1*$^{Cre/+}$;TgVegfr3 (heterozygous for *Tbx1*) embryos (Fig 6C–c') and in *Tbx1*$^{Cre/lacZ}$;TgVegfr3 (homozygous for *Tbx1*) embryos compared with controls (Fig 6E–e' and F), that is, they returned to wild-type levels. As *Tbx1*$^{Cre}$-driven recombination in the mouse brain is limited to ECs (shown in Fig 2), these results indicate that *Tbx1*-driven activation of the *Vegfr3* transgene in brain vessels is sufficient to compensate for *Tbx1* loss of function and restore normal brain vessel density in this mouse model.

Double immunostaining with anti-VEGFR2 and anti-GFP antibodies on brain sections of TgVegfr3;*Tbx1*$^{Cre/+}$ and *Tbx1*$^{Cre/+}$ embryos at E18.5 (n = 3 per genotype) did not reveal differences in the VEGFR2 signal intensity between the two genotypes (Fig S5B, C, H, and K).

In conclusion, the data from our loss and gain of function genetic experiments are consistent with the *Tbx1* → *Vegfr3* gene cascade that establishes the correct brain vessel density in the developing mouse brain. As full rescue of the brain vascular anomalies caused by the *Tbx1* mutation was achieved by activating *Vegfr3* exclusively in *Tbx1*-expressing ECs, we propose that critical TBX1 functions in brain angiogenesis are cell-autonomous.

## Discussion

In the mouse, the perineural vascular plexus (PNVP) begins to envelop the neural tube at E8.5. It forms by vasculogenesis through the coalescence of angioblasts that originate in the lateral plate mesoderm and paraxial mesoderm and migrate toward the neural tube in response to morphogenic gradients of VEGF (Hogan et al, 2004; Ruhrberg & Bautch, 2013; Puelles et al, 2019). The first brain vessels sprout from the PNVP and invade the hindbrain neuroepithelium around E9.5–E10.

In this study, we first addressed the question as to whether *Tbx1* is activated in EC progenitors (angioblasts) or in mature brain ECs. The results of the time course analysis of *Tbx1* cell fate mapping strongly suggest that the gene is first activated in *KDR*-expressing ECs in hindbrain blood vessels. What is the evidence for this? First, before the onset of brain vascularization (E9.5) and immediately afterward (E10.5), only a few GFP+ cells were observed in the PNVP. Second, in the head mesenchyme adjacent to the PNVP, GFP+ cells were KDR-negative. As migrating angioblasts activate KDR (Hogan et al, 2004), this suggests that GFP+ cells in the head mesenchyme are destined to contribute to nonvascular tissues and structures of the head. Third, at E10.5, isolated GFP+;KDR+ cells were observed in the hindbrain blood vessels, but we did not observe entire blood vessels or vessel networks that were GFP+ at this stage. The latter would be expected if *Tbx1* were activated in EC progenitors or in the PNVP before the onset of brain vascularization. After E11.5, the number of GFP+ cells in vessels within the neuroepithelium increased progressively in more rostral and dorsal brain regions, as

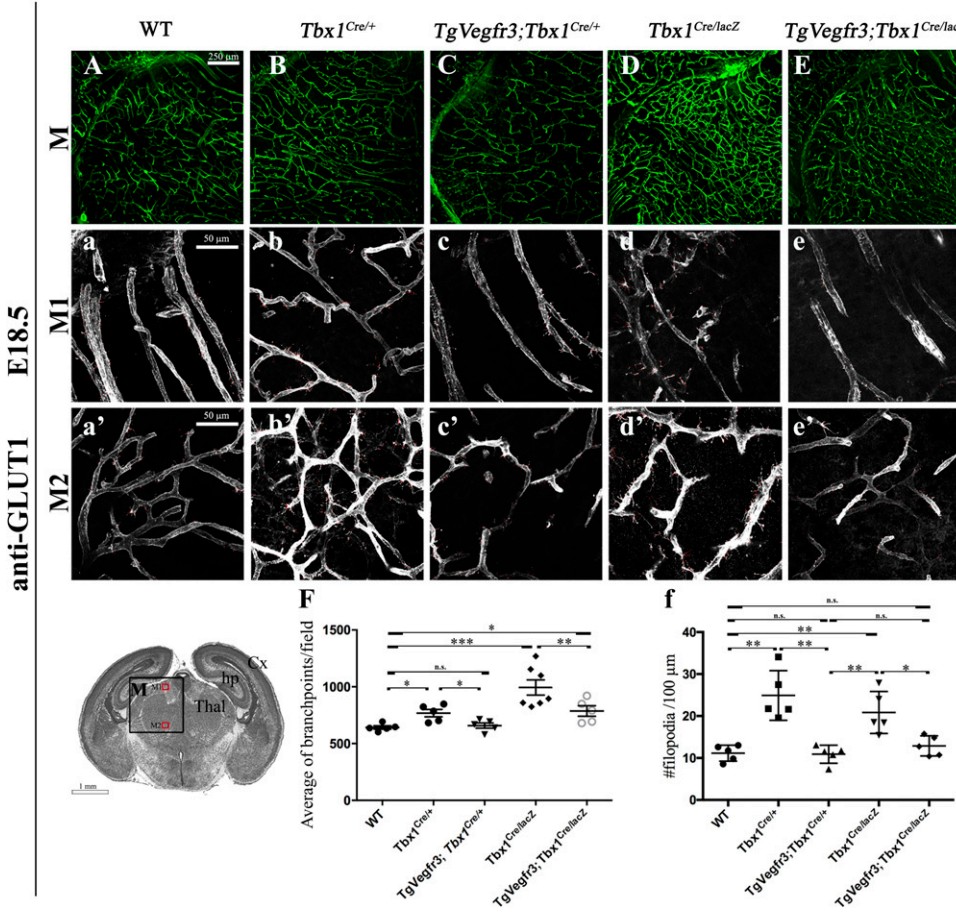

**Figure 6. *Vegfr3* overexpression rescues brain vessel abnormalities in *Tbx1* mutants.**
**(A, B, C, D, E)** Representative coronal brain sections (medial) of embryos at E18.5 immunostained for GLUT1 (green). **(F)** The cartoon indicates the position of the counting boxes (M1, M2), shown at high magnification in panels M1 (a–e) and M2 (a'–e'), that were used to quantify vessel branch point (F) and filopodial (f) density in embryos with the indicated genotypes. ***P-value < 0.001, **P-value < 0.01, *P-value < 0.05. Error bars ± SD. Abbreviations: Cx, cortex; hp, hippocampus; Thal, thalamus.

illustrated in the cartoon in Fig 1. Thus, the distribution of *Tbx1*-fated cells likely reflects a wave of *Tbx1* activation in differentiated ECs in the growing brain vascular network, rather than the deployment of *Tbx1*-expressing EC progenitors from the head mesenchyme. The fact that the brain vascular phenotype was only detectable in *Tbx1* mutants after E13.5 is consistent with the interpretation that critical TBX1 functions are in differentiated ECs and not in EC progenitors. In this study, we were not able to determine whether *Tbx1* is also activated in the pericyte lineage, which derives from mesodermal progenitors, like ECs. However, we have previously demonstrated that conditional deletion of *Tbx1* using the EC-specific Tie2Cre line is sufficient to recapitulate the hyperbranching phenotype (Cioffi et al, 2014), indicating that EC expression is critical.

By the end of embryogenesis, the GFP reporter driven by *Tbx1^Cre^* was activated in most brain regions, but compared with ventral brain structures (ventral forebrain, striatum, and hypothalamus), we observed a low contribution of GFP+ cells to blood vessels in the dorsal forebrain and midbrain, including the dorsal part of the cortex, hippocampus, and thalamus. Thus, for reasons unknown, *Tbx1* expression is not activated in the entire brain vascular network during embryogenesis, suggesting that there is heterogeneity in brain ECs. In contrast, VEGFR3-labeled vessels were evenly distributed throughout the brain. Thus, it is possible that in dorsal

brain regions, the regulatory function of TBX1 is performed by other transcription factors yet to be identified.

We found VEGFR3 to be expressed in most brain vessels between E13.5 and E18.5, which is in agreement with a study in which a *Vegfr3-Gap43-Venus* BAC transgenic mouse revealed pan-endothelial expression of the *Venus* reporter in brain vessels between E14.5 and E17.5 of murine embryogenesis and analogous expression of the endogenous VEGFR3 protein, identified by anti-VEGFR3 immunofluorescence (Watanabe et al, 2019). Thus, TBX1-GFP and VEGFR3 co-localize in a subpopulation of brain ECs from the timepoint in which the brain vessel hyperbranching phenotype was first identifiable in *Tbx1* mutant embryos (E15.5), suggesting that the copresence of the two proteins may be part of the pathogenetic mechanism. VEGFR3 expression is essential for angiogenesis in early mouse embryos (Dumont et al, 1998), but outside the CNS, it is down-regulated in systemic blood vessels when the lymphatic system begins to develop around E10.5, and by E16.5 its, expression is largely confined to lymphatic vessels, although it is reactivated in tumor blood vessels (Valtola et al, 1999; Tammela et al, 2008) and after injury (Paavonen et al, 2000). In contrast, in the brain parenchyma, which lacks lymphatic vessels, it continues to be expressed in brain microvessels throughout development (this study and Watanabe et al [2019]). Surprisingly, EC-specific inactivation of *Vegfr3* led to retinal and brain vessel hyperplasia

(Tammela et al, 2011), indicating that VEGF-C-VEGFR3 signaling can be pro- and antiangiogenic. VEGFR3 function has been most widely studied in the retina. Some studies have suggested that VEGFR3 does not play a primary role in angiogenesis, and that the phenotypic consequences of its inactivation upon this process are because of modulation of VEGFR2. For example, targeted inactivation of the tyrosine kinase or the ligand-binding domain of VEGFR3 did not affect angiogenesis in mouse embryos or yolk sacs (Zhang et al, 2010). As VEGFR3 and VEGFR2 form heterodimers, inactivation of the *Vegfr3* gene has been shown to affect signaling through the VEGFR2 receptor (Zarkada et al, 2015). Our study does not exclude this possibility because inactivation of TBX1 in cultured HUVECs and in brain ECs isolated from *Tbx1* mutants led to reduced expression of *Vegfr3* and increased expression of *Vegfr2* (Cioffi et al, 2014). In addition, TBX1 has been shown to regulate negatively *Vegfr2* gene expression in other contexts (Lania et al, 2015). Furthermore, in this study, the rescue of microtubule hyperbranching in *TBX1* knockdown ECs by forced *VEGFR3* expression was associated with reduced expression of *VEGFR2*. Thus, VEGFR2 signaling might function as one of the downstream effectors of the *TBX1*-*VEGFR3*–dependent modulation of vessel branching. In vitro studies in an appropriate microvascular EC model might reveal greater details about the nature of the TBX1-*Vegfr3* interaction in brain vessels and, potentially, the effect on brain EC behavior. This would require the investigation of additional molecules, including but not limited to *Vegfr2*, and the results would require in vivo validation.

Another intriguing possibility is that bimodal functions of VEGFR3 in ECs (pro- or antiangiogenic) depend upon local levels of Notch (Benedito et al, 2012). Several studies have contributed to the definition of an autoregulatory loop in which VEGF-C-VEGFR3 signaling activates Dll4-Notch, which promotes the conversion of endothelial tip cells into stalk cells, and subsequently suppression of the VEGF receptors, including VEGFR3 (Tammela et al, 2008, 2011; Jakobsson et al, 2009; Benedito et al, 2012), thereby preventing excessive angiogenesis. We have proposed that TBX1 operates upstream of this autoregulatory loop (Cioffi et al, 2014) because genetic inactivation of *Vegfr3* and *Dll4*, which are regulated by TBX1, and of *Tbx1* itself, all result in brain vessel hyperbranching (Suchting et al, 2007; Tammela et al, 2011; Cioffi et al, 2014), presumably due to the dominance of tip cells over stalk cells. The results of our current study sustain this hypothesis because reduced expression of *Vegfr3* enhances the *Tbx1* mutant phenotype, whereas increased expression suppresses it. In addition, *Vegfr3* and *Tbx1* are co-expressed in tip cells, consistent with an interaction that represses vessel branching.

Because TBX1 regulates *Vegfr3* expression at the transcriptional level (Chen et al, 2010), we hypothesize that this may be the mechanistic basis of the observed interactions, but is this enough? There are several points to consider: (i) the increased brain vessel density in *Tbx1*$^{-/-}$ embryos is not severe, thus TBX1 appears to have only a limited, modulatory/suppressor function over vessel branching; (ii) TBX1 positively regulates at least three genes that suppress angiogenesis, *Vegfr3*, *Dll4*, and *Unc5b* (Cioffi et al, 2014), the inactivation of any one of which leads to increased brain vessel density, similar to that observed in *Tbx1* mutants (Lu et al, 2004; Suchting et al, 2007; Tammela et al, 2011); (iii) a JAG1 peptide only partially rescued microtubule hyperbranching in TBX1-depleted

HUVECs (Cioffi et al, 2014), suggesting that Notch-independent pathways are involved. Thus, although it is possible that the loss of TBX1 perturbs a genetic circuit modulating vessel branching through multiple mechanisms, our data demonstrate that enhanced expression of *Vegfr3* is sufficient to rebalance or override it and re-establish normal branching. Thus, we conclude that *Vegfr3* is a critical target of TBX1 in brain microvessels.

## Materials and Methods

### Mouse lines and genotyping

Mouse studies were performed according to the animal protocol 257/2015-PR reviewed by the local IACUC committee and by the Italian Istituto Superiore di Sanità and approved by the Italian Ministero della Salute, according to Italian law and European guidelines. The following mouse lines were used: *Tbx1*$^{lacZ/+}$ (Lindsay et al, 2001), *Tbx1*$^{Cre/+}$ (Huynh et al, 2007), *Vegfr3*$^{flox/+}$ (Zarkada et al, 2015), *Vegfr3*$^{+/-}$ (Martucciello et al, 2020), *Rosa*$^{mTmG}$ (Muzumdar et al, 2007), and TgVegfr3 (Martucciello et al, 2020). Genotyping of mice was performed as in the original reports.

### Immunofluorescence on brain sections

Cryosections: E9.5, E10.5, E11.5 embryos were fixed in 4% PFA/1× PBS at 4°C overnight, washed in 1× PBS, and incubated for 12 h in serial dilutions of sucrose/1× PBS (10%, 20%, and 30% sucrose) at 4°C. Brains were then incubated for 2 h at 4°C in 50:50 vol/vol 30% sucrose/1× PBS/OCT before embedding in OCT. 10-$\mu m$ coronal sections were cut along the rostral-caudal brain axis on a cryotome. Alternatively, specimens were stored at −80°C. Experiments were performed on serial sections 200 $\mu m$ apart (five embryos/genotype unless stated otherwise). Sections were briefly microwaved to boiling point in 10 mM sodium citrate (pH 6.0) three times, for antigen enhancement, cooled, rinsed in 1× PBS, permeabilized with 0.1% Triton x-100, and blocked in 10% goat serum (GS)/1× PBS/0.1% Triton x-100 for 1 h at RT.

Thick brain sections: E13.5, E15.5, and E18.5 mouse brains were fixed in 4% PFA at 4°C overnight and subsequently embedded in 4% low melting agarose in 1× PBS. Serial thick coronal sections of 50 $\mu m$ (E13.5) or 100 $\mu m$ (E15.5, E18.5) were cut on a vibratome at 4°C. Brains (fixed) that were not used immediately were stored at 4°C in 0.0025% NaNO$_3$.

Immunofluorescence was performed on serial coronal sections along the rostral-caudal brain axis using the following antibodies: mouse monoclonal anti-GLUT1 (ab40084; Abcam), chicken polyclonal anti-GFP (ab13970; Abcam), anti-mouse monoclonal VEGFR3/FLT4 (clone AFL4, Cat. no. 14-5988-85; eBioscience), rabbit polyclonal anti-VEGFR3/FLT4 (#36398; Elabscience), and rat anti-mouse monoclonal VEGFR2/KDR (Avas 12α1, Cat. no. 550549; BD Pharmingen). Secondary antibodies used were goat anti-mouse Alexa Fluor 594 and 488, goat anti-rabbit Alexa Fluor 488 and 594, goat anti-rat Alexa Fluor 594, and goat anti-chicken Alexa Fluor 488. Sections were then incubated with primary antibodies overnight at 4°C in the same blocking solution reducing the GS to 5%, rinsed, and

incubated in secondary antibodies for 1 h 30 min at RT. Fluorescence was observed with an epifluorescence microscope (Leica DMI6000B, acquisition software LAS AF 2.6; Nikon Confocal Microscope A1, mounted on Nikon ECLIPSE Ti, acquisition software NiS element, or Leica Thunder Imaging System [Leica Microsystems, Wetzlar]) equipped with a LEICA DFC9000 GTG camera and a Lumencor fluorescence LED light source. For the latter, a ×63 oil immersion objective was used to acquire Z-slice images or a ×100 oil immersion objective for experiments involving pericyte markers. Small-volume computational clearing was used to remove the background signal derived from out-of-focus blur. Images were digitally documented with a camera and computer-processed using Adobe Photoshop version 6 for Windows.

## Quantitative analysis of vascular anomalies

All quantitative analyses were performed on three brain sections per embryo in rostral, medial, and caudal postions that correspond visually to the following stereotaxic levels (Paxinos & Franklin, 2012). Rostral: bregma 0.74 mm; medial: bregma −2.54 mm; caudal: bregma −4.84 mm, where R: encompasses the dorsal and ventral sub-pallium; M: the dorsal and ventral thalamus; and C: the dorsal and ventral midbrain.

### *Vessel branch point and filopodia counts*
We analyzed a minimum of five embryos per genotype in all experiments. Flattened images were digitally reconstructed from confocal *z* stacks of 2.5 $\mu$m, representing three different brain regions (rostral, medial, and caudal). For each flattened image, we manually counted all the branch points in a single field per section and thus three fields per embryo. The area counted per field was 1.444 mm$^2$, and the total length of the vascular network per field was measured. For filopodia, we counted all the filopodia in two sub-fields per section (rostral, medial, and caudal) for a total of six fields per embryo. The area counted per field was 0.045 mm$^2$, and counts are expressed as the mean number of filopodia per 100 $\mu$m of vessel length. Counts were performed using the ImageJ cell counter plugin.

## Statistical analysis

The statistical analysis of the data pertaining to the brain vessel density was performed using a likelihood ratio test for negative binomial generalized linear models. We first calculated the mean number of branch points/field/embryo. We then calculated the mean for the group (embryos/genotype). The latter value was used for the statistical analysis. For the analysis of data pertaining to filopodia, we performed the Kruskal–Wallis test followed by the unpaired Mann–Whitney U test.

## Cell manipulations

HUVECs (Lonza) and bEND5 (Sigma-Aldrich), brain ECs of BALB/c mice were transfected by Lipofectamine 2000 (Invitrogen).

For *TBX1* or *VEGFR3* knockdown (KD) in HUVECs, RNA interference was performed using a commercial siRNA for *TBX1* or *VEGFR* (ON-TARGETplus SMARTpool; Thermo Fisher Scientific) (40 nM) or a control (non-targeted) siRNA (Thermo Fisher Scientific) as

previously described (Cioffi et al, 2014). mRNA expression was evaluated by qreal-time PCR. For *Vegfr3* overexpression, HUVECs were transfected with 3 $\mu$g of pcDNA3 plasmid vector containing a full-length mouse *Vegfr3* cDNA (GeneCopoeia) or with an empty vector (Martucciello et al, 2020). 24 h after transfection, cells were collected and processed for Matrigel assay.

For *Vegfr3* overexpression in *Tbx1*$^{KD}$ ECs, 24 h after transfection of HUVECs with a *TBX1* or control siRNAs, cells were transfected with pcDNA3-*Vegfr3*. 24 h after the second transfection, cells were collected and processed for Matrigel assays.

bEND5 cells were cultured in DMEM supplemented with 0.2 mM L-glutamine, 50 units/ml penicillin, 50 mg/ml streptomycin, and 20% (vol/vol) fetal bovine serum (GIBCO). For *Vegfr3* overexpression, cells were transfected with 3 $\mu$g of the pcDNA3 plasmid vector containing a full-length mouse *Vegfr3* cDNA (GeneCopoeia); for *TBX1* overexpression, cells were transfected with 3 $\mu$g of pcDNA3 plasmid vector containing a full-length human *TBX1* cDNA or an with empty vector. *Tbx1*, *Vegfr3*, *Vegfr2*, and *Nrarp* mRNA expressions were evaluated by qreal-time PCR.

## Matrigel assays

200 $\mu$l of Matrigel (BD Bioscience) was plated onto chilled 15-mm wells and incubated at 37°C for 30 min, as per the manufacturer's instructions. HUVECs in six-well plates, previously transfected as described above, were trypsinized and counted. 1.5 × 10$^5$ treated cells in EGM-2 media (Lonza) were added to each well containing Matrigel. After 16 h at 37°C, the formation of microtubules was analyzed using an Olympus CKX41 Image Analyzer. The quantification of microtubule branch points was performed after dividing each large image into nine sub-images. The number of branch points was calculated as the sum of counts made in all nine sub-images. Quantitative analysis was performed using the ImageJ software.

# Supplementary Information

# Acknowledgements

We are grateful for the invaluable support provided by the Integrated Microscopy Core, the Animal Facility, and in particular Lucia Mele, at the Institute of Genetic and Biophysics "ABT"/CNR, Naples. We thank Drs. Claudia Angelini and Annamaria Carissimo of the Istituto Applicazioni del Calcolo, CNR, Naples, for assistance with the statistical analysis and Dr. Giuseppina Divisato of the Dipartment Molecular Medicine and Medical Biotechnology, University of Naples Federico II, for assistance with microscopy. *Vegfr3*$^{flox/+}$ mice were generously provided by Dr. Kari Alitalo. The study was supported by grants from the Fondation Leducq Transatlantic Network of Excellence in Cardiovascular Research, 15CVD01 (to E Illingworth), from the Jerome Lejeune Foundation, 1685 (to E Illingworth), and from the Italian Ministry of Health #20179J2P9J (to E Illingworth). MG Turturo was supported by a doctoral fellowship from the European School of Molecular Medicine (SEMM).

## Author Contributions

S Cioffi: data curation, formal analysis, validation, investigation, visualization, and methodology.

G Flore: data curation, formal analysis, validation, investigation, visualization, and methodology.

S Martucciello: data curation, formal analysis, validation, investigation, visualization, and methodology.

M Bilio: data curation, formal analysis, investigation, visualization, and methodology.

MG Turturo: data curation, formal analysis, validation, investigation, visualization, and methodology.

E Illingworth: conceptualization, supervision, funding acquisition, and writing—original draft, review, and editing.

## Conflict of Interest Statement

The authors declare that they have no conflict of interest.

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
