## [Reviewer comments · Life Science Alliance]

VEGFR3 modulates brain microvessel branching in a mouse model of 22q11.2 deletion syndrome

Sara Cioffi, Gemma Flore, Stefania Martucciello, Marchesa Bilio, Maria Giuseppina Turturo, and Elizabeth Illingworth
DOI: <https://doi.org/10.26508/lsa.202101308>

Corresponding author(s): Elizabeth Illingworth, University of Salerno

Review Timeline:

Submission Date:	2021-11-20
Editorial Decision:	2021-12-22
Revision Received:	2022-08-08
Editorial Decision:	2022-09-06
Revision Received:	2022-09-15
Editorial Decision:	2022-09-16
Revision Received:	2022-09-23
Accepted:	2022-09-23

Scientific Editor: Novella Guidi

Transaction Report:

December 22, 2021

Re: Life Science Alliance manuscript #LSA-2021-01308-T

Prof. Elizabeth Anne Illingworth
university of salerno
chemistry and biology
Via Giovanni Paolo II, 132
fisciano, SA 84084
Italy

Dear Dr. Illingworth,

Thank you for submitting your manuscript entitled "VEGFR3 modulates brain microvessel branching in a mouse model of 22q11.2 deletion syndrome" to Life Science Alliance. The manuscript was assessed by expert reviewers, whose comments are appended to this letter. We, thus, encourage you to submit a revised version of the manuscript back to LSA that responds to all of the reviewers' points.

Thank you for this interesting contribution to Life Science Alliance. We are looking forward to receiving your revised manuscript.

Sincerely,

B. MANUSCRIPT ORGANIZATION AND FORMATTING:

Reviewer #1 (Comments to the Authors (Required)):

The data here reported extend previous observations published in 2014 by this group on the role of Tbx1 in regulating brain vascularization during mouse development. In particular they focused on the earlier stages of development establishing that Tbx1 is expressed in ECs but not in their precursors. Interestingly, the endothelial expression of Tbx1 is dynamically expressed along the development a limited at specific areas of the brain. Because VEGFR3 null mice showed a phenotype similar to that observed in Tbx1^{+/-} mutant with an increased vascular density, the authors investigated the putative interaction between Tbx1 and VEGFR3. By exploiting mouse genetic models they demonstrated that haploinsufficiency of Vegfr3 increased the vascular defect observed in Tbx1^{+/-} mutant. On the basis of in vitro experiments showing that the VEGFR3 overexpression partially rescued the Tbx1 silencing effect in an in vitro angiogenic model, the authors crossed Tbx1^{Cre/+} mice with mice carrying a single copy of a Cre-activatable Vegfr3 transgene and showed that Vegfr3 overexpression rescued the vascular effect of Tbx1 mutant. The data here shown are interesting but do not fully support the authors' conclusions.

CRITICISMS

The immunofluorescence analysis along embryo development does not support the final conclusion that Tbx1 is not expressed in endothelial precursors. More EC progenitor markers are required to support this hypothesis (e.g. CD31, MECA-20, MECA-32, Flt-1, c-kit, Sca-1, VE-cadherin, CD14).

I agree that the in vivo analyses here proposed are not resolutive and allow the role of other genetic interactions (e.g. Tbx1/VEGFR2). Therefore, more solid in vitro experiments could better support the model proposed. A first relevant criticism is the use of HUVEC that are far from the biological context of brain microvasculature. The isolation of brain EC is feasible (the authors used these cells in the paper published in 2014) and the in vivo models set up by the authors could be an important cellular source to dissect the interaction between VEGFR3 and Tbx1.

The use of brain EC from WT mice and the in vitro genetic manipulation of VEGFR2 might be also necessary to put in the real context the putative compensatory role occurring between VEGFR2 and VEGFR3, which are respectively up- and down regulated in Tbx1 mutant, according to the paper on Human Molecular Genetics.

Finally, it is mandatory to understand how Tbx1 regulates the expression of VEGFR3. For instance, does Tbx1 is a direct repressor of Vegf3?

In the discussion it should be important to discuss the zonal effect/expression of Tbx1 in ECs.

Reviewer #2 (Comments to the Authors (Required)):

In this paper, Cioffi et al investigate the role of Tbx-1 mediated VEGFR3 expression in developmental angiogenesis in the mouse brain. They use some elegant mouse models to suggest that VEGFR3 regulates early angiogenesis in the brain and is in turn regulated by the transcription factor Tbx-1. In addition, the work is novel in providing data regarding the previously unknown role of VEGFR3 in vascularising the developing brain. However, in concentrating exclusively on Tbx-1 and VEGFR3, the study is rather limited in scope and lacking in mechanistic detail. Indeed, the authors concede that VEGFR2, the isoform most commonly associated with angiogenesis, may also play a role, especially since previous work by this group indicates Tbx1 can negatively regulate VEGFR2 expression. It would be interesting therefore to investigate what happens to VEGFR2 expression in the mouse models used in this study as this would perhaps provide a more complete picture.

Reviewer #3 (Comments to the Authors (Required)):

In their manuscript Cioffi et al. explore the link between the transcription factor TBX1 and VEGFR3 during embryonic brain vascularization. They utilize a genetic lineage tracing mouse model (Tbx1^{Cre};Rosam^{TmG}) to map the fate of Tbx1⁺ ECs in the brain and show that only ECs express TBX1 in the parenchyma and that GFP⁺ ECs express VEGFR3. They then explore the genetic interaction of Tbx1 and Vegfr3 by analyzing double haploinsufficient Tbx1^{+/-};Vegfr3^{+/-} mice, as well as by

overexpressing VEGFR3 in a TBX1-mutant background. My main concern is that the authors use the expression of GFP as a readout for TBX1 localization. It is most likely that the cell localization of GFP is different than TBX1 protein, as the GFP expression seen here is typical of most EC-specific drivers crossbred with mTmG (uniform and strong expression throughout the whole cell; also good filopodia staining). In addition, the GFP half-life may be longer than TBX1, and not accurately representing the TBX1 protein expression pattern. It would be therefore necessary to confirm the accuracy of the Tbx1Cre;RosamTmG mouse as a tool to monitor TBX1 expression, either by IHC using an anti-TBX1 antibody or in situ hybridization for Tbx1.

Overall, the methods and analysis of this manuscript do not support adequately the authors' conclusions and the manuscript needs to undergo major revision prior to publication.

Major concerns

- 1) In Fig 1-2, why is only a subset of ECs express TBX1? Also, the expression of TBX1 seems to go up from E9.5 onwards. It would be useful to provide a quantification of %GFP positive KDR/GLUT blood vessels at different timepoints. The same holds for the co-localization of GFP with VEGFR3 shown later: the authors need to provide quantitative data of various timepoints analyzed, to conclude with respect to the predicted highest interaction timepoint between those proteins/genes.
- 2) In some of the high-mag pictures provided (e.g. Fig 3MN, 3OP), it looks as if some GFP+ cells surround the VEGFR3+ cells. Have the authors performed double IF for pericyte markers to exclude the possibility that some GFP+ cells are in fact mural cells?
- 3) In Figure 4, it is hard to appreciate the differences in filopodia projections. First the baseline (WT) numbers are different in the quantifications shown in f and f' (~10 and 20 for WT in f and f' respectively). Is there an explanation for this difference? If there is a plausible explanation (e.g. different genetic backgrounds) the authors could report % in changes vs WT, to allow for comparisons between mutants. Currently it is not clear to me whether double haploinsufficiency for Vegfr3 and Tbx1 produces more filopodia compared to Vegfr3 haploinsufficiency alone (~25/100um vs. 35/100um for Vegfr3+/-), and no conclusions for genetic interactions can be drawn.
- 4) The results in Figure 5 suggest that VEGFR3 may not be the only gene downstream of TBX1 regulating branching in the in vitro microtubule assay. Therefore, Vegfr3 overexpression should not be described as a "rescue", rather than an "improvement". Since the authors have previously shown that Notch activation partially improves microtubule hyperbranching in Tbx1-KO HUVECs, they could combine VEGFR3 overexpression with Notch activation and test whether this treatment could provide a full rescue. Also, the authors should discuss why the in vivo gain of function model provides a full rescue compared with the in vitro assay.
- 5) In figure 6, did the authors confirm overexpression in the brain blood vessels of their mice and lack of ectopic expression of VEGFR3?
- 6) In the discussion the authors write: suggesting that Notch-independent pathways contribute to the pathological process. Here we investigated the TBX1-VEGF-C/VEGFR3 axis. VEGFR3 signaling is not Notch independent, as loss of VEGFR3 signaling in blood ECs results to loss of Notch signaling (Tammela et al, 2011, Zarkada et al, 2015). This section needs re-writing to be in line with the literature.
- 7) Later in the discussion, the authors write: There are no published studies of VEGFR3 function specifically in brain vessels. This is inaccurate as Tammela et al reported on hindbrain vascularization in Vegfr3iEKO mouse embryos (2011).
- 8) Our study does not exclude this possibility as inactivation of Tbx1 in cultured HUVECs and in brain ECs isolated from Tbx1 mutants led to increased expression of Vegfr2 (Cioffi et al., 2014). Vegfr3 loss of function results to VEGFR2 decrease at the protein and mRNA levels (Zarkada et al, 2015). It is therefore possible that VEGFR2 is upregulated following TBX1 loss-of function because of VEGFR3 downregulation. This should be discussed.

Minor concerns

- 1) Please add page numbers and figure numbers.
- 2) It would be useful to add brief descriptions of the markers that are used for IF, since some readers may not be familiar with what structures they are supposed to stain (i.e. KDR, GLUT).

Response to the reviewers

Reviewer 1.

Point 1. *The immunofluorescence analysis along embryo development does not support the final conclusion that Tbx1 is not expressed in endothelial precursors. More EC progenitor markers are required to support this hypothesis.*

Response:

The conclusion that *Tbx1* is not expressed in EC progenitors is based on the results of the fate mapping of *Tbx1*-expressing cells and not on cell phenotype (Results first section, Discussion second para.).

In the revised manuscript, Discussion para. 1 cites the origin of brain EC progenitors in the mouse, and highlights the results that led to the conclusion that *Tbx1* is activated in mature brain ECs rather than EC progenitors, namely, i) the low number of GFP+ cells in the PNVP before and even after the onset of brain vascularization; ii) the lack of KDR+;GFP+ cells in the head mesenchyme where KDR-expressing angioblasts differentiate and migrate towards the PNVP, iii) the observation of isolated GFP+ cells in hindbrain vessels, but not whole vessels or vessel networks expressing GFP. Finally, in the Discussion, page 11, para. 2 now reads: ... *Thus, the distribution of Tbx1-fated cells likely reflects a wave of Tbx1 activation in mature ECs in the growing brain vascular network, rather than the deployment of Tbx1-expressing EC progenitors from the head mesenchyme.*

We have found no evidence in the literature of local sources of EC progenitors in the brain parenchyma or brain vessels during mouse embryonic development.

Point 2

(a) *... the in vivo analyses here proposed are not resolute and allow the role of other genetic interactions (e.g Tbx1/VEGFR2). Therefore, more solid in vitro experiments could better support the model proposed.*

Response:

This manuscript focuses on the *Tbx1*-*Vegfr3* interaction. Of course there will be additional interactions at play, but we do not have the reagents and resources to address multiple ones, and they would be outside of the scope of this study, in any case. We agree with the reviewer when he/she says that *VEGFR2* may be involved. In fact, we say exactly this in the Discussion, page 13, para.1. It is not clear what *in vitro* experiments the reviewer has in mind but in the revised manuscript we present new data on *VEGFR2*, Results, page 10, para.2, Supplementary Figures 4 and 5. See also responses to point 3 from this reviewer and point 1 from Reviewer 2. Whether the modest changes of expression observed are sufficient to cause the phenotype is very difficult to say and it would require an entire new project to investigate this with the appropriate genetic tools. Furthermore, *Tbx1* regulates many genes and several of them have affect angiogenesis (e.g. Notch and Netrin pathways), to simply focus on one gene may not provide a balanced view.

(b) *..... A first relevant criticism is the use of HUVEC that are far from the biological context of brain microvasculature. The isolation of brain EC is feasible (the authors used these cells in the paper published in 2014) and the in vivo models set up by the authors could be a important cellular source to dissect the interaction between VEGFR3 and Tbx1.*

Response:

Unfortunately, we cannot manipulate freshly isolated mouse brain ECs (WT or *Tbx1*^{+/-}) because we are unable to maintain them in culture. In the study by Cioffi et al., 2014, they were used immediately after isolation for gene expression studies. For some experiments requested by the reviewers we have used a commercial mouse brain EC line, bEND5, but in this cell line and in

primary mBECs *Tbx1* expression is barely detectable, thereby precluding *in vitro* studies of *Tbx1*-*Vegfr3* interaction, except in the context of *Tbx1* over-expression (see response to point 3). We respectfully note that we have a small colony of mice (mouse protocols in Italy are very restrictive), therefore to redo all the crosses and collect endothelial cells would take us at least one year of work.

Point 3. The use of brain EC from WT mice and the in vitro genetic manipulation of VEGFR2 might be also necessary to put in the real context the putative compensatory role occurring between VEGFR2 and VEGFR3, which are respectively up- and down regulated in Tbx1 mutants.

Response:

The current study does not intend to dissect putative compensatory roles between VEGFR2 and VEGFR3 (see also our response to the previous point). There is quite a lot of literature about this (some of it cited in this manuscript) that indicates an elevated level of complexity likely involving the Notch pathway and probably other signalling pathways. Within the scope of this study, is the effect of altered *Tbx1* and *Vegfr3* gene dosage on vessel branching. In the revised manuscript, new *in vitro* experiments (cited in response to point 2) are described in the Results, page 10, para.2, where we now state... "we tested whether *Vegfr2* dysregulation also occurs in a mouse brain endothelial cell line, bEND5. We found that overexpression of *Tbx1* and *Vegfr3* in these cells were both associated with modest but significant down regulation of *Vegfr2* expression (Supplementary Figure S4-A-B). Analogously, in HUVECs *VEGFR2* expression increased after *TBX1* knockdown and returned to control levels of expression in cells co-transfected with *TBX1* siRNA and pcDNA3-*Vegfr3* (Supplementary Figure S4-C). Thus, *Vegfr2* dysregulation may contribute to the branching phenotype observed in cultured cells."

Point 4. It is mandatory to understand how Tbx1 regulates the expression of Vegfr3. For instance, is Tbx1 a direct repressor of Vegfr3?

Response:

TBX1 is a direct, positive regulator of *Vegfr3* gene expression. Published data from our group demonstrate that TBX1 up regulates *Vegfr3* in ECs, including HUVECs, LECs and BECs (Chen et al. 2010; Cioffi et al. 2014; Martucciello et al. 2020); all three studies showed that *Tbx1* mutation leads to reduced expression of *Vegfr3* in ECs. Furthermore, Chen et al. 2010 showed by ChIP and transgenic analyses that TBX1 activates *Vegfr3* by binding to an enhancer element in the endogenous *Vegfr3* gene. We do not know whether this enhancer is active in brain ECs. However, obtaining mechanistic insights into how TBX1 regulates *Vegfr3* expression in brain ECs is currently not possible because the commercially available anti-TBX1 antibodies are not suitable for ChIP.

Point 5. In the discussion it is important to discuss the zonal effect/expression of Tbx1 in ECs.

Response:

We have added the following to the Discussion, page 12, para.2... "By the end of embryogenesis, the GFP reporter driven by *Tbx1*^{Cre} was activated in most brain regions, but compared to ventral brain structures (ventral forebrain, striatum, hypothalamus) we observed a low contribution of GFP+ cells to blood vessels in the dorsal forebrain and midbrain, including the dorsal part of the cortex, hippocampus, and thalamus. Thus, for reasons unknown, *Tbx1* expression is not activated in the entire brain vascular network during embryogenesis, suggesting that there is heterogeneity in brain ECs"

Reviewer 2.

In concentrating exclusively on Tbx1 and VEGFR3, the study is rather limited in scope and lacking in mechanistic detail.... It would be interesting therefore to investigate what happens to VEGFR2 expression in the mouse models used in this study...

Response:

Please also see our response to points 2a, 3 and 4 from Reviewer 1.

We have analyzed *in vivo* VEGFR2 protein expression, Results, page 11, para.2 now states... "We also performed anti-VEGFR2 immunostaining on brain sections of TgVegfr3;Tbx1^{Cre/+} and Tbx1^{Cre/+} embryos at E18.5. This did not show any appreciable difference in VEGFR2 expression between the two genotypes (Supplementary Figure S5b, c, h, k). We also performed real time PCR on RNA extracted from whole brains of Tbx1^{Cre/+};TgVegfr3 vs Tbx1^{Cre/+} embryos at E18.5, n = 3 per genotype. This too revealed no significant difference in Vegfr2 expression between the two genotypes (data not shown). While it is difficult to exclude that VEGFR2 may function as a downstream effector of both TBX1 and VEGFR3, we think that there may be a contribution by this receptor but it is unlikely that it will be the only effector; TBX1 regulates many genes involved in angiogenesis, and VEGFR3, which is not a transcription factor, may have indirect effects on many genes, too.

Reviewer 3.

Introductory paragraph: My main concern is that the authors use the expression of GFP as a readout for TBX1 localization. It is most likely that the cell localization of GFP is different than TBX1 protein, as the GFP expression seen here is typical of most EC-specific drivers crossbred with mTmG (uniform and strong expression throughout the whole cell; also good filopodia staining)

Response:

We respectfully disagree with the reviewer on this point. Tbx1Cre-induced GFP expression is used to identify Tbx1-expressing cells and their descendants, not to establish TBX1 protein sub-cellular localization, which is nuclear. GFP encoded by the the Rosa^{mTmG} reporter allele is targeted to the cytoplasmic membrane, so it is ideal to establish whether the same cell also expresses other membrane proteins such as VEGFR3. We have made changes to the text in order to avoid any possible confusion or interpretational ambiguity. In particular, Results, section 2 has a new title, "*Endothelial tip cells are potential sites of Tbx1 and Vegfr3 interaction*". Results, page 8, para.2 we have added the following... "Thus, GFP+; VEGFR3+ ECs are potential sites of Tbx1-Vegfr3 interaction".

Point 1

(a) *In Fig 1-2, why is only a subset of ECs express TBX1?*

Response:

See response to Reviewer 1, point 5 which poses a similar question.

(b) *Also, the expression of TBX1 seems to go up from E9.5 onwards.*

Response:

Indeed, expression is expanded but we do not know whether it increases (GFP expression reflects Tbx1 cell lineage). As development progresses there is a rapid increase in the number of labelled cells because Tbx1-expressing cells and their descendents express GFP.

(c) *It would be useful to provide a quantification of % GFP positive KDR/GLUT blood vessels at different timepoints.*

Response:

The % GFP+;GLUT1+ brain vessels in E18.5 embryos (79.1%) has been reported (Cioffi et al., 2014).

In this study, the goal of the double immunostaining was to determine whether *Tbx1* was activated in non-endothelial cell types, i.e., in KDR-negative or GLUT1-negative cells. This was not the case. The expansion of GFP+ cells is dependent upon stage and region. It would be extremely complicated to draw a quantitative picture and it is unclear to us what additional information we would gain compared to the extensive qualitative information provided.

d) The same holds for the co-localization of GFP with VEGFR3 shown later: the authors need to provide quantitative data of various timepoints analyzed, to conclude with respect to the predicted highest interaction timepoint between those proteins/genes.

Response:

The suggested quantification would be useful if GFP expression reflected *Tbx1* expression, but as this is not the case, it would not be very informative, especially at later stages of development when most brain vessels are GFP+ (Supplementary Figure 1), and most, if not all GFP+ vessels are also VEGFR3+. In the Results, page 8, para. 2 we now define GFP+;VEGFR3+ cells as potential sites of genetic interaction.

*e) It would be necessary to confirm the accuracy of the *Tbx1*Cre;Rosa^{mTmG} mouse as a tool to monitor TBX1 expression, either by IHC using an anti-TBX1 antibody or in situ hybridization for *Tbx1*.*

Response:

Our *Tbx1*^{Cre} line has been used widely in the literature to monitor *Tbx1*-expressing cells and their descendants (cell fate). It does not monitor gene expression directly. In the brain, we have also used a *Tbx1*^{lacZ} knock-in allele, which monitors *Tbx1* gene expression (Paylor, 2006) and it was detected in brain vessels (the Cre and lacZ reporters use the identical insertion point into the *Tbx1* gene). Available anti-TBX1 antibodies do not work after E9.5 for unknown reasons. *In situ* hybridization and RNA scope are both theoretically possible, but they do not label efficiently ECs, which form a thin mono-layer of cells.

Point 2. Have the authors performed double IF for pericyte markers to exclude the possibility that some GFP+ cells are in fact mural cells?

Response:

We thank the reviewer for raising this point. We have performed triple IF on brain sections of *Tbx1*^{Cre/+}; *Rosa*^{mTmG} embryos at E18.5 using anti-PDGFR β to label pericytes, anti-GLUT1 to label brain ECs and anti-GFP to label *Tbx1*-fated cells. Representative optical sections are shown in Supplementary Figure 2) and the results described on page 7, last para. The IF revealed the co-localization of all three signals in most brain vessels, and moreover they co-localized in the vessel structures in randomly selected single optical sections within each z stack (n=3 for each genotype). Thus, at the maximum magnification that we can achieve (x100 objective), we were unable to distinguish between EC-derived signals (GLUT1+) and pericyte-derived signals (PDGFR β +). Thus, we are not able to provide a definitive answer to the reviewers question, Discussion, page 12, para.1 states.... "In this study we were not able to determine whether *Tbx1* is also activated in the pericyte lineage, which derives from mesodermal progenitors, like ECs. However, we have previously demonstrated that conditional deletion of *Tbx1* using the EC-specific Tie2Cre line is sufficient to recapitulate the hyperbranching phenotype, (Cioffi et al., 2014), indicating that EC

expression is critical". It is quite likely that *Tbx1* is activated in the pericyte lineage, which is mesodermal in origin, as *Tbx1* is activated in the anterior mesoderm at E7.5 (Huynh et al., 2007).

Point 3.

(a) *In Figure 4, it is hard to appreciate the differences in filopodia projections. First the baseline (WT) numbers are different in the quantifications shown in f and f' (~10 and 20 for WT in f and f' respectively). Is there an explanation for this difference?*

Response:

The mutants analyzed in Figure 4 are presented in separate bar charts because *Vegfr3*^{+/-} mice are derived from a different cross (*Vegfr3*^{fllox} and *Vegfr3*^{+/-} mice cannot be generated in the same cross). *Vegfr3*^{+/-} mice are presented here in order to show the *Vegfr3* haploinsufficiency phenotype, not as a control for Figure 4F and 4f.

(b) *The authors could report % in changes vs WT, to allow for comparisons between mutants. Currently it is not clear to me whether double haploinsufficiency for Vegfr3 and Tbx1 produces more filopodia compared to Vegfr3 haploinsufficiency alone (~25/100um vs. 35/100um for Vegfr3+/-), and no conclusions for genetic interactions can be drawn.*

Response:

We have added percentage values to the dot plots in Figure 4 (4f, 4f'). However, direct comparison between *Tbx1*^{Cre/+; Vegfr3}^{fllox/+} mice and *Vegfr3*^{+/-} mice cannot be made because the former are conditional *Vegfr3* heterozygotes, while the latter are germline *Vegfr3* heterozygotes.

Point 4.

(a) *Vegfr3 overexpression should not be described as a "rescue", rather than an "improvement"*

Response:

In the revised manuscript, we talk about improved microtubule branching in *TBX1*-knockdown ECs that over-express *Vegfr3*. We prefer to retain the term "rescue" in the *in vivo* context, i.e., when a treatment (pharmacological or genetic) fully restores the normal phenotype.

(b) *Since the authors have previously shown that Notch activation partially improves microtubule hyperbranching in Tbx1-KO HUVECs, they could combine VEGFR3 overexpression with Notch activation and test whether this treatment could provide a full rescue.*

Response:

We have performed this experiment using the JAG1 peptide reported by Cioffi, 2014 and we are providing the data for the reviewer to see (attached at the end of this letter). Unfortunately, we were only able to perform two experimental replicates. We cannot procure more Matrigel in a reasonable time. Current predictions for delivery are January 2023, but recent experience tells us that this is not reliable. Of course, colleagues are unwilling to assist, given this market situation.

The results obtained with two biological replicates showed that while both JAG1 and pcDNA-Vegfr3 alone partially rescued microtubule hyperbranching in *Tbx1*-KO HUVECs, when they were combined, the level of rescue improved marginally.

We do not wish to include in a manuscript results derived from only two independent experiments. We hope that the reviewer and the Editors will understand our position and consider the *in vivo* results far more significant than the *in vitro* one.

(c) *Discuss why rescue is partial in the in vitro study and full in the in vivo study*

Response:

We have added the following to the Results, page 11, para.1, "As *Tbx1*^{Cre}-driven recombination in the mouse brain is limited to ECs (as shown in Figure 2), these results indicate that *Tbx1*-driven activation of the *Vegfr3* transgene in brain vessels is sufficient to compensate for *Tbx1* loss of function and restore normal brain vessel density in this mouse model. This was not the case for cultured ECs, where forced *Vegfr3* expression only partially rescued microtubule hyperbranching in *Tbx1*-knockdown HUVECs. This difference might be due to the lack of one or more critical factors required for VEGFR3 function in these cells, or to the use of a less than optimal dose of pcDNA-Vegfr3.

Point 5. In figure 6, did the authors confirm overexpression in the brain blood vessels of their mice and lack of ectopic expression of VEGFR3?

Response:

The *Vegfr3* transgene is not tagged, therefore we cannot distinguish between endogenous and transgenic *Vegfr3* expression in *Vegfr3*-expressing tissues. Appropriate *Tbx1*^{Cre}-induced activation of the transgene (in *Tbx1*-expressing tissues) and functional testing have been reported (Martucciello, 2020). We have now performed double IF with anti-VEGFR3 and anti-GFP antibodies on brain sections of *Tbx1*^{Cre/+};*Rosa*^{mTmG} vs *Tbx1*^{Cre/+};*Rosa*^{mTmG} ; TgVegfr3 embryos. This did not reveal significant differences in VEGFR3 signal intensity between the two genotypes, as described in the Results, page 10, para. 3 and shown in (Supplementary Figure S5a, d, g, j). This is not altogether surprising because this transgenic line was selected in order to minimize the risk of producing high levels of VEGFR3 expression that might cause embryonic lethality. Please note that the endogenous *Vegfr3* gene is expressed in all brain vessels, it co-localizes with GLUT1, as shown in Supplementary Figure 1, and as reported by Watanabe et al., 2019.

With regard to ectopic expression, Figures 1 and 2 show that *Tbx1*^{Cre}-activated expression of GFP is limited to brain ECs, which is the only *Tbx1* expression domain in the brain. We therefore expect *Tbx1*^{Cre}-activation of TgVegfr3 to be similarly brain EC-specific as the transgene is activated by Cre recombination.

Point 6. In the discussion the authors write: suggesting that Notch-independent pathways contribute to the pathological process. Here we investigated the TBX1-VEGF-C/VEGFR3 axis. VEGFR3 signaling is not Notch independent, as loss of VEGFR3 signaling in blood ECs results to loss of Notch signaling (Tammela et al, 2011, Zarkada et al, 2015). This section needs re-writing to be in line with the literature.

Response:

We apologize for this error. The two phrases have been removed.

Point 7. There are no published studies of VEGFR3 function specifically in brain vessels. This is inaccurate as Tammela et al reported on hindbrain vascularization in Vegfr3iEKO mouse embryos (2011).

Response:

We thank the reviewer for pointing this out. We intended no published studies dedicated to elucidating *Vegfr3* function in brain vessels. The Discussion, page 13, para. 1 now states, "VEGFR3 function has been most widely studied in the retina....". The study by Tammela et al. 2011 is cited multiple times in the manuscript.

Point 8. *Vegfr3 loss of function results to VEGFR2 decrease at the protein and mRNA levels (Zarkada et al, 2015). It is therefore possible that VEGFR2 is upregulated following TBX1 loss-of function because of VEGFR3 downregulation. This should be discussed.*

Response:

The reviewer meant to say *Vegfr3 loss of function results in increased VEGFR2 at the protein and mRNA levels* (Figure 3K, 3L Zarkada et al. 2015). In the revised manuscript, we have cited this study in the Discussion, page 13, para. 1), and we then state, " Our study does not exclude this possibility because inactivation of *Tbx1* in cultured HUVECs and in brain ECs isolated from *Tbx1* mutants led to reduced expression of *Vegfr3* and increased expression of *Vegfr2* (Cioffi et al., 2014). In addition, *Tbx1* has been shown to regulate negatively *Vegfr2* gene expression in other contexts (Lania et al. 2016). Furthermore, in this study, the rescue of microtubule hyperbranching in *TBX1* knockdown ECs by forced *VEGFR3* expression was associated with reduced expression of *VEGFR2*. Thus, VEGFR2 signalling might function as one of the downstream effectors of the *Tbx1-Vegfr3*-dependent modulation of vessel branching.

See response to Reviewer 1, points 2, 3, 4, and Reviewer 2 for new experiments involving VEGFR2.

Minor concerns

1) Please add page numbers and figure numbers.

Response: This has been done.

2) It would be useful to add brief descriptions of the markers that are used for IF, since some readers may not be familiar with what structures they are supposed to stain (i.e. KDR, GLUT).

Response: This information has been added.

A) Quantitative analysis of microtubule branchpoints following the treatments shown in panel B. a) Expression of Nrarp by qRT-PCR. B) Images of representative endothelial microtubular networks. Abbreviations: Ctr, control, EV, empty vector, scr, scrambled, pep, peptide, n.s. not significant, *** P value <0.001, ** P value <0.01, * P value <0.05.

A

B

a

September 6, 2022

Re: Life Science Alliance manuscript #LSA-2021-01308-TR

Prof. Elizabeth Anne Illingworth
university of salerno
chemistry and biology
Via Giovanni Paolo II, 132
fisciano, SA 84084
Italy

Dear Dr. Illingworth,

Thank you for submitting your revised manuscript entitled "VEGFR3 modulates brain microvessel branching in a mouse model of 22q11.2 deletion syndrome" to Life Science Alliance. The manuscript has been seen by the original reviewers whose comments are appended below. While the reviewers continue to be overall positive about the work in terms of its suitability for Life Science Alliance, some important issues remain. Please address Reviewer 1 concerns by discussion and address Reviewer 3 last points, including point 2 if doable.

Our general policy is that papers are considered through only one revision cycle; however, given that the suggested changes are relatively minor, we are open to one additional short round of revision. Please note that I will expect to make a final decision without additional reviewer input upon resubmission.

Please submit the final revision within one month, along with a letter that includes a point by point response to the remaining reviewer comments.

To upload the revised version of your manuscript, please log in to your account: <https://lsa.msubmit.net/cgi-bin/main.plex>
You will be guided to complete the submission of your revised manuscript and to fill in all necessary information.

B. MANUSCRIPT ORGANIZATION AND FORMATTING:

Sincerely,

Reviewer #1 (Comments to the Authors (Required)):

The MS has been improved but in my opinion several limits linger. In particular the use of HUVEC to dissecting tissue-specific vascular processes is acceptable but the at least some of the most relevant data obtained have to be confirmed on microvascular ECs, possibly isolated from the specific tissues. The authors claim the impossibility to pass their brain ECs. I underline that many companies commercialize human microvascular ECs and some of them also from brain. By silencing TBX1 the authors could better demonstrate the effect of TBX1 on VEGFR3, and biological parameters of ECs related to angiogenesis (migration, adhesion, morphogenesis). About the use of bEND cell line I'm a little bit suspicious, because they have been originated by immortalization of murine ECs with mT oncogene, which constitutively activates src and deeply influences signaling pathways.

Reviewer #2 (Comments to the Authors (Required)):

The authors have satisfactorily addressed my concerns. I have no other issues

Reviewer #3 (Comments to the Authors (Required)):

The revised manuscript is improved, and I am satisfied with the changes that the authors made. I only have a few more comments and requests:

- 1) Please state explicitly that you are performing a lineage tracing study with the Tbx1Cre;mTmG line.
- 2) It is worrying that the authors cannot demonstrate increased VEGFR3 expression in the TgVegfr3 embryos (Point 5), since this experiment is central to the manuscript's conclusions. Would it be possible to lyse brain tissue from Tbx1Cre/+;RosamTmG and Tbx1Cre/+;RosamTmG ;TgVegfr3 embryos and perform RTqPCR or western blot analysis instead of IHC?
- 3) Please mark supplemental figures with their numbers

Reviewer #1 (Comments to the Authors (Required)):

1) The MS has been improved but in my opinion several limits linger. In particular the use of HUVEC to dissecting tissue-specific vascular processes is acceptable but the at least some of the most relevant data obtained have to be confirmed on microvascular ECs, possibly isolated from the specific tissues. The authors claim the impossibility to pass their brain ECs. I underline that many companies commercialize human microvascular ECs and some of them also from brain. By silencing TBX1 the authors could better demonstrate the effect of TBX1 on VEGFR3, and biological parameters of ECs related to angiogenesis (migration, adhesion, morphogenesis).

Response:

Our approach is to use *in vivo* studies as the gold standard. We therefore performed some preliminary *in vitro* studies before embarking on costly and time consuming *in vivo* studies. Our motive was, as stated in the manuscript, to gain insights into the likely success of achieving phenotypic rescue *in vivo*. However, we recognize the value of *in vitro* studies in an appropriate microvascular cell model. In the revised manuscript, Discussion page 10, para. 1 we have added the following: " *In vitro* studies in an appropriate microvascular endothelial cell model might reveal greater details about the nature of the TBX1-VEGFR3 interaction in brain vessels and, potentially, the effect on brain EC behavior. This would require the investigation of additional molecules, including but not limited to *Vegfr2* and the results would require *in vivo* validation".

As to sourcing additional primary brain microvascular EC cell lines that might express higher levels of *Tbx1*, this would delay the completion of this study. With current delivery times it would take up to a year to receive the cell lines and characterize them, with no guarantee of being able to perform the *Tbx1* silencing experiments. Furthermore, with regard to the use of non-brain microvascular endothelial cells to investigate an interaction that takes place in brain ECs. I would be very cautious, given the evidence (cited in the manuscript) that *Tbx1* and *Vegfr3* function differently inside and outside the CNS. Again, any results obtained would require *in vivo* validation.

Reviewer #3 (Comments to the Authors (Required)):

The revised manuscript is improved, and I am satisfied with the changes that the authors made. I only have a few more comments and requests:

1) Please state explicitly that you are performing a lineage tracing study with *Tbx1Cre;mTmG*.

Response:

Results page 4, para. 1 now states " To this end, we crossed *Tbx1^{Cre/+}* mice with *Rosa^{mTmG}* reporter mice (Muzumdar et al., 2007) and evaluated the distribution of *Tbx1*-expressing cells and their descendants (lineage tracing) in serial coronal.....

2) It is worrying that the authors cannot demonstrate increased VEGFR3 expression in the *TgVegfr3* embryos (Point 5), since this experiment is central to the manuscript's conclusions. Would it be possible to lyse brain tissue from *Tbx1Cre/+;RosamTmG* and *Tbx1Cre/+;RosamTmG;TgVegfr3* embryos and perform RTqPCR or western blot analysis instead of IHC?

Response:

We have performed this experiment in *Tbx1^{Cre/+};Rosa^{mTmG}* and *Tbx1^{Cre/+};Rosa^{mTmG};TgVegfr3* embryos at E18.5. RTqPCR performed on whole brain tissue revealed increased expression of *Vegfr3* in mutant (*Tbx1^{Cre/+};Rosa^{mTmG};TgVegfr3*) compared to control (*Tbx1^{Cre/+};Rosa^{mTmG}*) brains (P=0.009, P=0.010). The new results are described on page 7, para. 2 and the data, normalized to GAPDH and PECAM1 expression, have been added to Supplementary Fig 5 (m). We hope that these new results will satisfy the legitimate concern of the reviewer and future readers.

3) Please mark supplemental figures with their numbers

Response: The Assistant Editor requests that all figures files are without numbers.

September 16, 2022

RE: Life Science Alliance Manuscript #LSA-2021-01308-TRR

Prof. Elizabeth Anne Illingworth
university of salerno
chemistry and biology
Via Giovanni Paolo II, 132
fisciano, SA 84084
Italy

Dear Dr. Illingworth,

Thank you for submitting your revised manuscript entitled "VEGFR3 modulates brain microvessel branching in a mouse model of 22q11.2 deletion syndrome". We would be happy to publish your paper in Life Science Alliance pending final revisions necessary to meet our formatting guidelines.

- please add the Twitter handle of your host institute/organization as well as your own or/and one of the authors in our system
- please add a conflict of interest statement and the author contributions to the main manuscript text
- please consult our manuscript preparation guidelines <https://www.life-science-alliance.org/manuscript-prep> and make sure your manuscript sections are in the correct order

Figure Check:

- please expand on your figure legend panels to be clearer in your figure legends
- Figure 1: scale bars needed for C and D

A. FINAL FILES:

B. MANUSCRIPT ORGANIZATION AND FORMATTING:

Sincerely,

September 23, 2022

RE: Life Science Alliance Manuscript #LSA-2021-01308-TRRR

Prof. Elizabeth Anne Illingworth
university of salerno
chemistry and biology
Via Giovanni Paolo II, 132
fisciano, SA 84084
Italy

Dear Dr. Illingworth,

Thank you for submitting your Research Article entitled "VEGFR3 modulates brain microvessel branching in a mouse model of 22q11.2 deletion syndrome". It is a pleasure to let you know that your manuscript is now accepted for publication in Life Science Alliance. Congratulations on this interesting work.

DISTRIBUTION OF MATERIALS:

Again, congratulations on a very nice paper. I hope you found the review process to be constructive and are pleased with how the manuscript was handled editorially. We look forward to future exciting submissions from your lab.

Sincerely,
